# The Characteristics of PM$_{2.5}$ and PM$_{10}$ and Elemental Carbon Air Pollution in Sevastopol, Crimean Peninsula

Alla V. Varenik

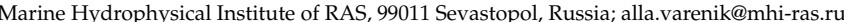

Marine Hydrophysical Institute of RAS, 99011 Sevastopol, Russia; alla.varenik@mhi-ras.ru

**Abstract:** In most cities of the world, air pollution reaches critical levels. The air masses circulating over the Crimean Peninsula bring a significant amount of mineral dust, which contains soil particles, emissions from industrial enterprises, gases, etc. The purpose of this research is to study the processes and the factors influencing atmospheric pollution in Sevastopol (Crimea). Air pollutant concentration data, including elemental carbon, nutrients (inorganic fixed nitrogen, inorganic fixed phosphorus and silicon), PM$_{10}$, and PM$_{2.5}$, were collected during this research. Samples were collected at the station that is located at a distance from sources of pollution (background station). Our study has shown that even at the background site the daily-averaged concentrations of PM$_{10}$ and PM$_{2.5}$ particles in the atmosphere of Sevastopol reach and even exceed the maximum permissible concentrations in the case of dust transported from deserts. Values of the daily-averaged concentrations of microparticles have exceeded the European maximum permissible concentration (MPC) values in 17 cases for PM$_{2.5}$ particles and in 6 cases for PM$_{10}$. The impact of both local sources and long-distance atmospheric transport depends on weather conditions. Concentrations of elemental carbon in air samples have never exceeded the maximum allowed by regulations concentration limits during our research. However, the elemental carbon concentration in air samples collected near highways with a traffic intensity of approximately 500–1000 cars per hour has exceeded the background values by 30–50 times.

**Keywords:** PM$_{10}$; PM$_{2.5}$; dust transport; nutrients; elemental carbon





## 1. Introduction

Atmospheric pollution is the global problem of our time. In the last decades, scientists discovered that climate change is also due to air pollution. Atmospheric pollution can be caused by different factors. One of the most significant is transport, including auto transport and air transport (passenger or business).

Polluted air is transported over long distances, and it affects people's health and their environments. The indicators that are commonly used to characterize particle matters (PM), which are of great importance for human health, include the mass concentration of particles with a diameter smaller than 10 microns (PM$_{10}$) and smaller than 2.5 microns (PM$_{2.5}$). PM$_{2.5}$, which are often called fine suspended particles, also include ultrafine particles with a diameter below 0.1 microns. PM with a diameter from 0.1 to 1 microns can be in the air for many weeks and, accordingly, be a subject of transboundary transport. Microparticles can be of natural (dust transport, soil erosion) and anthropogenic origin [1,2]. Internal combustion engines, combustion of various types of fuels (coal, brown coal, heavy oil, biomass) in stationary heat sources (boiler houses), construction, many types of production (especially cement, ceramics, bricks, smelting), and trans-shipment of bulk cargo are the main sources of suspended particles in the urban atmosphere.

The most common chemical components of PM include sulfates, nitrates, ammonia, other inorganic ions, such as sodium, potassium, calcium, magnesium and chloride ions, organic and elemental carbon (soot), minerals of the earth's crust, metals (including vana-

dium, cadmium, copper, nickel, and zinc), etc. The PM also contains biological components, such as allergens and microorganisms.

In 2016, the World Health Organization (WHO) published a report on the content of $PM_{10}$ and $PM_{2.5}$ in the atmosphere of 2975 cities around the world (www.who.int/phe/ heath_topics/outdoorair/databases/AAP-database, accessed on 14 June 2022). This report indicates that the quality of atmosphere depends to some extent on the socio-economic state of the country. Concentrations of suspended particles are higher in cities of Asian countries and the Eastern Mediterranean as compared to developed countries. However, such significant factors as the proximity of cities to arid territories and traffic load have not been taken into account. Droughts and intensive exploitation of pastures lead to reduction in vegetation at large areas of arid zones in Asia and in Africa and make them the source of dust particles. Dust from such territories is transported for hundreds and thousands of kilometers [3]. Therefore, a high level of atmospheric pollution with dust particles of various diameters in a number of cities may be associated with the transport of dust from these territories [4,5]. In summer, the amount of precipitation may decrease by 40% [6], which may lead to accumulation of dust in the atmosphere.

The WHO in the "Air Quality Standards for fine particles" (EN 12341:2014) and the European Union in the form of Directives on the quality of air (Directive 2008/50/EC and document EN 12341:2014) attributed suspended particles $PM_{10}$ and $PM_{2.5}$ to be the most significant factors of air pollution affecting the health of the population. Therefore, in the modern world there is an acute problem of informing people about the level of air pollution.

The analysis of soot of natural and anthropogenic origin is especially important. Almost every process of burning emits soot. The highest concentration is observed in urban and suburban areas. However, because of the ubiquity of emission sources and due to the long residence time of particles in the atmosphere, soot in the air can be observed also in remote areas, such as the Arctic [7–9]. Soot particles are the second most important factor in global warming after carbon dioxide [10,11]. Accurate determination of the amount of soot particles is important for climate monitoring and forecasting based on atmospheric data assimilation [12].

In Europe and in the USA, studies of the distribution of suspended particles in the atmosphere have been carried out since the end of the twentieth century, and active development of such studies is currently taking place in Asian countries [13,14]. In China, for example, the weather forecast is always accompanied by information on the value of the $PM_{2.5}$ index. This information is extremely necessary since air pollution in many megapolises of China has reached critical values.

Taking into account the pandemic situation in 2020, one can expect that a reduction in automobile and air traffic should lead to decreasing air pollution. Indeed, Ekici et al. [15] made a quantitative comparison of aircraft engine emissions caused by domestic and international commercial flights before and after the COVID-19 pandemic in Turkey. They have shown a significant decrease in the amount of air pollutants.

For example, the updated global exposure estimates of $PM_{2.5}$ in different regions of the world [16] demonstrate further advances in characterizing global population exposure to ambient air pollution. The global coverage in this research allows for estimating concentrations of air pollutants in areas without extensive ground monitoring, including, for example, rural areas with large emissions from households using coal as fuel. Using these estimates in combination with model simulations of chemical transport can provide information on sector-specific contributions to air quality management.

However, to obtain more correct information, monitoring of ambient air pollution must be in place and comprehensive. Management of air quality is a very effective tool for assessment of air pollution on different scales—national (macro), city (medium), and local [17]. The successful implementation of the management of air quality plans in different regions depends on the efficiency of its key components, e.g., aim, monitoring network, emission inventory, air quality modeling, etc.

In the Russian Federation, approximately half of its population lives in cities, where air pollution levels are above MPC [18]. However, automated monitoring of $PM_{10}$ and $PM_{2.5}$ is currently organized only in large cities, such as Moscow, St. Petersburg, Sochi, Kazan, and others [19]. Compared to other countries, the surveillance network is not sufficiently developed and informative (http://aqicn.org/city/beijing/—The World Air Quality Index, accessed on 14 June 2022). Even less data is available on the composition of $PM_{10}$ and $PM_{2.5}$, although these particles are excellent adsorbents of various pollutants, such as trace elements and radioactive isotopes, complex organic substances, etc. [20]. This is especially true for microparticles $PM_{10}$ and $PM_{2.5}$, which is soot of natural (fires) and anthropogenic (products of fuel combustion and tire wear) origin.

In the Black Sea region, studies on the chemical composition and quantity of aerosols in the atmosphere have been rarely carried out, as compared to, for example, the Mediterranean Sea region [21,22]. As a rare example, researchers [22] have carried out studies of aerosols in the Blue Bay area of the Black Sea on the transect from Gelendzhik to the open sea, as well as at several coastal stations. It has been discovered that terrigenous sources are located at a short distance, and variations in the wind can completely change the elemental composition of aerosol samples during the synoptic period.

The chemical composition of aerosols in the eastern Black Sea was studied at a station located in the mountains of Turkey, 50 km from the Black Sea coast in 2011–2013 [23]. The authors concluded that there were no seasonal variations in the concentration of anthropogenic elements in aerosols. Metallurgical plants located in the study region were the main sources of anthropogenic aerosol in the atmosphere. There were also particles of soil and secondary sulfate in the atmosphere, transported over long distances by air masses.

In Sevastopol, measurements of the particle size distribution in the atmosphere have been carried out sporadically. According to published data, the highest values of the $PM_{2.5}$ concentration in ambient air have been observed for weather conditions with winds of the eastern direction, which indicates a terrigenous source of the particles to the air. $PM_{10}$ and $PM_{2.5}$ concentrations values have typically exceeded MPC values near roads [24,25].

This study represents results of long-term investigations of the $PM_{2.5}$, $PM_{10}$, and elemental carbon concentration levels in the atmosphere of Sevastopol city in Crimea. Air samples were collected at the background meteorological station. In order to compare the pollution level in different conditions, some air samples were collected in parallel near highways with a traffic intensity of approximately 500–1000 cars per hour.

## 2. Materials and Methods

### 2.1. Nutrients in Atmospheric Precipitations

To assess the input of nutrients with atmospheric precipitations in Sevastopol, samples of wet and bulk (wet + dry) atmospheric depositions were analyzed for the content of inorganic nitrogen, phosphates, and silicon. From 2004 to the present, samples were collected after each case of precipitation in two samplers: a wet-only sampler, which opened with the beginning of precipitation and closed immediately after it, and a permanently open sampler to assess the effect of dust input on the concentration of nutrients in precipitation. Samples were collected at the Sevastopol meteorological station according to Manual on Air Pollution Control [26]. This manual governs the monitoring of atmospheric pollution in cities at the regional and the background levels, methods of chemical analysis of concentrations of harmful substances in the atmosphere, methods of collection, and processing and statistical analysis of the results of observations. The methods in the manual are mandatory for all organizations to monitor atmospheric pollution; analyze air samples, precipitation, and snow for the content of harmful substances; collect, process, and analyze information; and compile generalized information about the air quality of cities at the regional and the background levels. The sampling site location and equipment are shown in Figure 1.

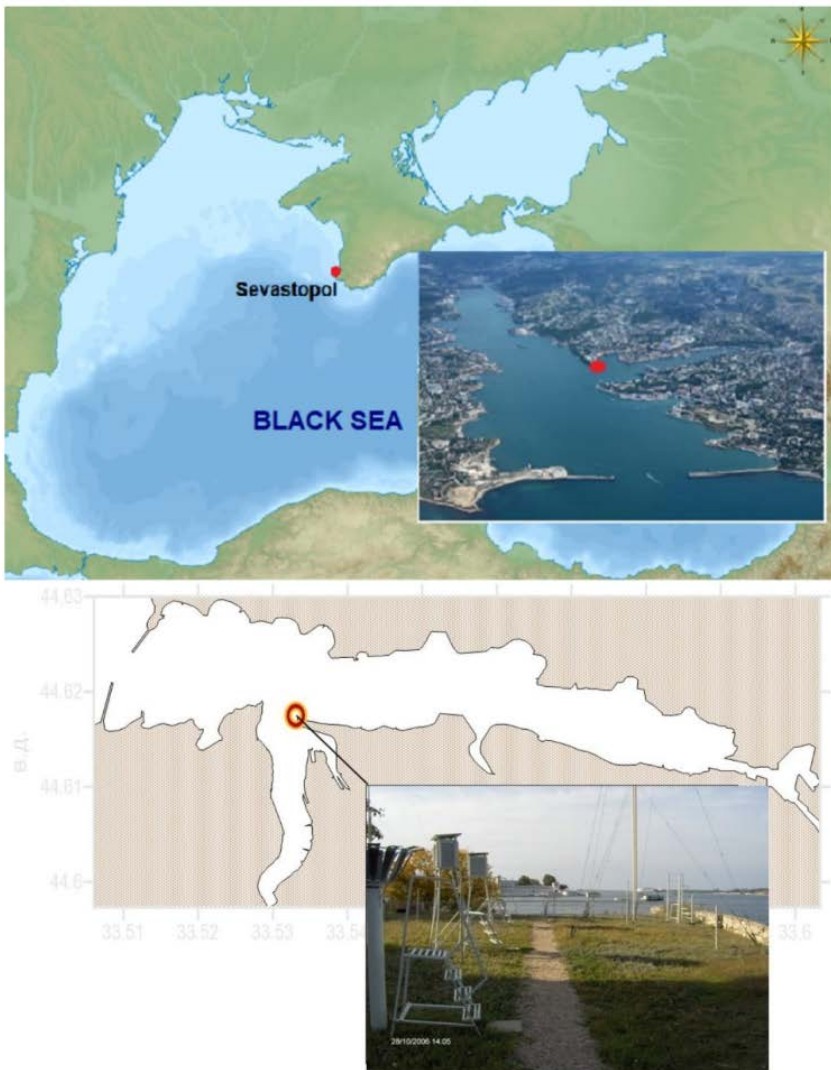

**Figure 1.** Sampling site.

The Sevastopol meteorological station has been located at the junction of the southern and northern bays, on the Pavlovsky Cape since 1909. In this place, for more than a century, round-the-clock monitoring of air temperature and humidity and wind direction and speed as an amount of precipitation has been conducted, which has allowed for the collection of the first and most important portions of precipitation. Taking into account that the station is located at a distance from sources of pollution, we consider it as a background sampling site.

Collected samples were analyzed photometrically at the Marine Hydrophysical Institute RAS (Sevastopol, Crimea). Ammonium was determined photometrically on a spectrophotometer (type KFK-3-01, Zagorsky Optical and Mechanical Plant, Sergiev Posad), using a modified Sadgi–Solorzano method [27]. Sodium nitroprusside was used as a catalyst for the reaction. As a result of the phenol-hypochlorite reaction, an intensely colored blue compound, indophenol, was formed. The maximum intensity of coloring was achieved within 6 h. Keeping samples and standards with added reagents in a thermostat at a temperature of 40–50 °C allowed for reducing the time of full development of coloring to 30–60 min. The method provided reliable results in the range of concentration from 0.1 to 15.0 $\mu$mol L$^{-1}$. Statistical processing of the results revealed that the minimum of ammonia concentrations determined by this method was at the level of 0.03–0.05 $\mu$mol L$^{-1}$. In the laboratory, 10 mL of each sample were placed in plastic tubes. Then, 0.2 mL of phenol solution was added and (after mixing) 0.2 mL of hypochlorite solution was added. Samples

and standards of the scale with added reagents were placed in a thermostat adjusted to 48–50 °C for 30 min. After cooling for 30 min in a dark place, the optical density of the colored solutions was measured in a cuvette of 5 cm, with a spectrophotometer at a wavelength of 648 nm.

To determine the nitrate–nitrite concentration, we used a Skalar_SAN++ continuous flow analyzer. The automated procedure for the determination of nitrate and nitrite was based on the cadmium reduction method; the sample was buffered at pH 8.2 and then passed through a column containing granulated copper–cadmium to reduce the nitrate to nitrite. The nitrite (originally present plus reduced nitrate) was determined by diazotizing with sulfanilamide and coupling with N-(1-naphthyl)ethylene diamine dihydrochloride to form a highly colored azo dye, which was measured at 540 nm. The automated procedure for the determination of nitrite was based on the following reaction: the diazonium compounds formed by diazotizing of sulfanilamide by nitrite in water under acid conditions was coupled with N-(1-naphthyl)ethylenediamine dihydrochloride to produce a reddish-purple color, which was measured at 540 nm. The method error was $\pm 0.20\ \mu M$ [28].

The concentration of silicon was determined based on formation of a blue silico-molybdenum complex. The method error was $\pm 6\%$. The determining of the inorganic phosphorus concentration was carried out by the method based on the formation of a blue phosphoromolybdenum complex. The method error was $\pm 10\%$. As a result, an array of data on concentrations in samples for each of the samplers was obtained. The cases of multiple excess of the content of nutrients in precipitations collected in an open sampler over concentrations in a wet-only sampler were determined.

### 2.2. $PM_{2.5}$ and $PM_{10}$

Air samples to assess $PM_{10}$ and $PM_{2.5}$ concentrations were collected with a dust meter "Atmas" (Russia, https://ntm.ru/products/150/8342, accessed on 14 June 2022) every day since February, 2020. The analyzer (Figure 2) was equipped with an impactor, with replaceable nozzles for fractional separation of suspended aerosol particles ($PM_{10}$, $PM_{2.5}$).

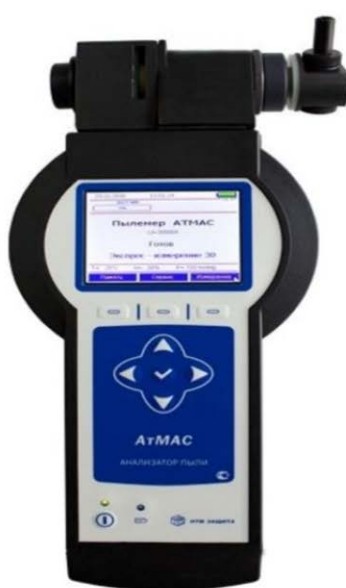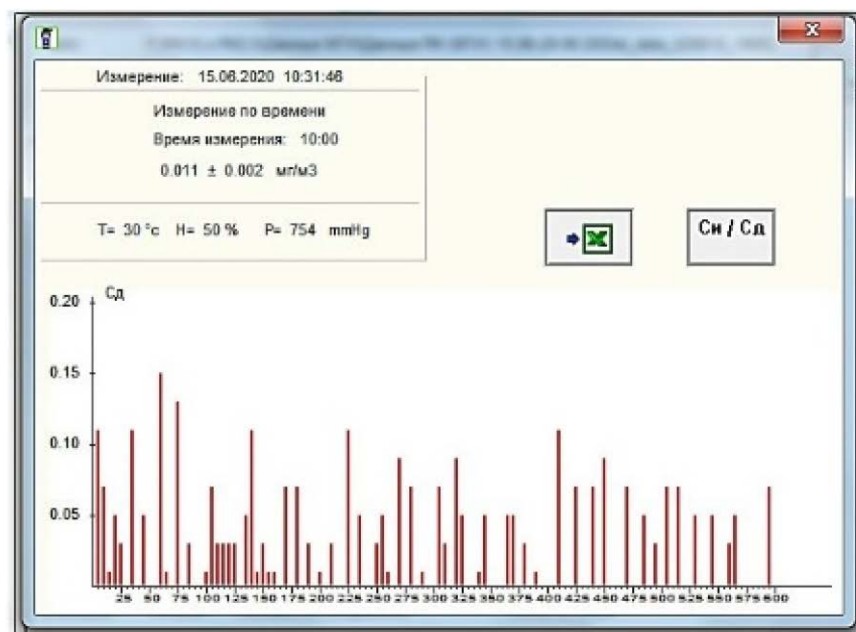

**Figure 2.** Dust meter "Atmas".

This dust meter could be used for a sanitary and hygienic technological inspection of the air environment of premises and used for work in the field. The dust meter directly measured the mass concentration. There was no need to adjust the conversion factor

for different dust composition. The dust meter operation principle was based on the piezoelectric measurement method, the essence of which was to measure the natural frequency of the piezoelectric element during the deposition of aerosol particles on its surface. When the air was pumped through the analyzer, the particles in the air sample entered the corona discharge field created by the electrode, where they received an electric charge and were deposited on the surface of the piezoelectric element. When particles were deposited on the surface of the piezoelectric element, the frequency of its oscillations changed, which was proportional to the mass of the settled dust.

The measurement range of the mass concentration of dust was 0.1–150 mg m$^{-3}$. The error in the concentration range from 0.1 to 20 mg m$^{-3}$ was equal to 20%. If the dust concentration was more than 20 mg m$^{-3}$, it was necessary to use a special diluent cartridge, while the error was $\pm$ 25%. Single values of the concentration of PM were recorded every 5 s.

Concentrations of microparticles in the atmosphere of Sevastopol were measured according to RD 52.04.186-89—Manual on Air Pollution Control [26] and EMEP [29] three times a day: at 10 a.m., 2 p.m. and 6 p.m.; the duration of exposure and the averaging period of PM$_{2.5}$ and PM$_{10}$ concentrations was 30 min (individual samples). The average concentration of these values was considered as the daily-averaged concentration.

### 2.3. Elemental Carbon in the Air

The method was based on collection of elemental carbon particles on the filter and further photometric determination of the carbon mass concentration in the suspension according to manual [30]. This guidance was intended for laboratories performing measurements in the area of atmospheric pollution monitoring. The range of elemental carbon concentrations in the atmosphere according to this manual was 0.03–1.8 mg m$^{-3}$, with a measurement accuracy of 18%.

To determine the concentration of elemental carbon, the air was aspirated through a filter in a holder with a flow rate of 20–40 dm$^3$ min$^{-1}$ for 30 min. Samples were collected at the windward side at a height of 1.5 m from the ground surface. After sampling, the filter was removed from the holder, folded with the dusty side inside, placed in a plastic bag, and delivered to a laboratory for analysis. Filters with selected samples were dried in a desiccator over anhydrous calcium chloride for 2 h. At low concentrations of elemental carbon in the atmosphere, it was allowed to determine the daily-averaged concentrations by taking at least four individual samples per filter during the day. The dried filter with the sample was placed along the tube wall with tweezers. Then, 5 cm$^3$ of dimethyl sulfoxide were added to the tube (the solvent should be poured directly onto the filter material), while the filter material gradually dissolved, and aerosol particles passed into a suspended state. To process the sample with an ultrasound, the tube with suspension was fixed on a tripod and placed in an ultrasonic bath filled with distilled water and then treated with ultrasound for 60 min. Then, all samples were analyzed photometrically. The ultrasound-treated samples were transferred to a cuvette, and optical density measurements were carried out at a wavelength of 400 nm on the spectrophotometer (type KFK-3-01, Russia). The sampling of the air to obtain data of elemental carbon concentration in Sevastopol started in October 2021 and remains in progress. To compare data obtained on the background station (Figure 1), we collected air samples near highways with the traffic intensity of approximately 500–1000 cars per hour and more.

## 3. Results and Discussions

### 3.1. PM$_{2.5}$ and PM$_{10}$ Concentration Measurements

During our research, more than 900 individual (duration of exposure is 30 min) values and approximately 300 daily-averaged values of PM$_{2.5}$ and PM$_{10}$ concentrations in the air were obtained. Some statistical characteristics of PM concentrations are in the Table 1.

**Table 1.** Statistical characteristics of PM daily-averaged concentrations in the atmosphere in Sevastopol.

|  | $PM_{10}$, mg m$^{-3}$ | $PM_{2.5}$, mg m$^{-3}$ |
| --- | --- | --- |
| Max | 0.114 | 0.057 |
| Min | 0.00 | 0.00 |
| St.Dev. | 0.013 | 0.009 |
| Average | 0.019 | 0.014 |
| Total number | 912 | 903 |

Time series plots of $PM_{10}$ and $PM_{2.5}$ concentrations in the atmosphere in Sevastopol are shown in Figure 3.

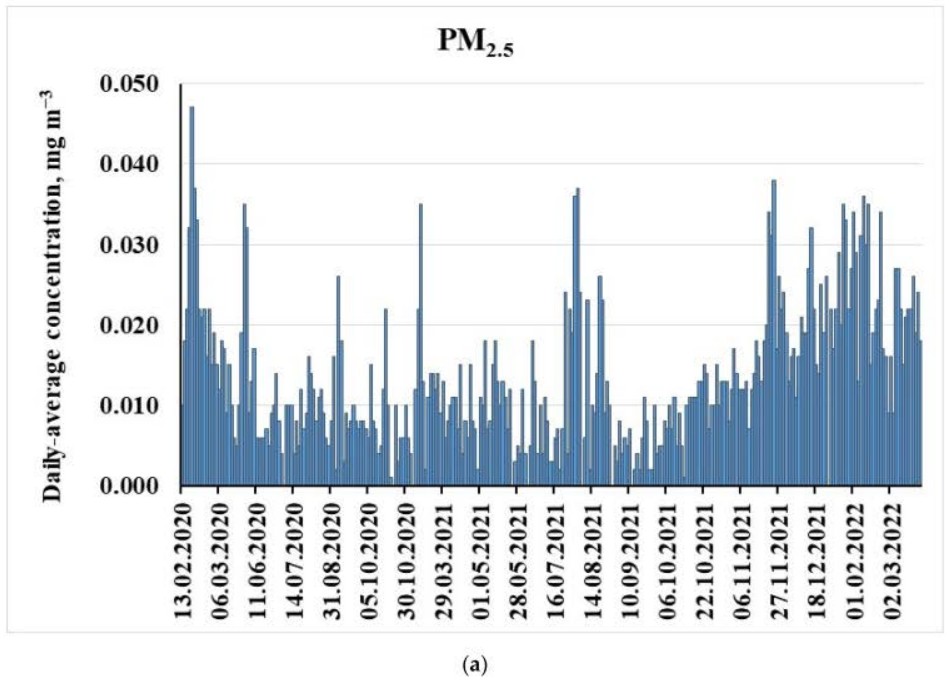

(a)

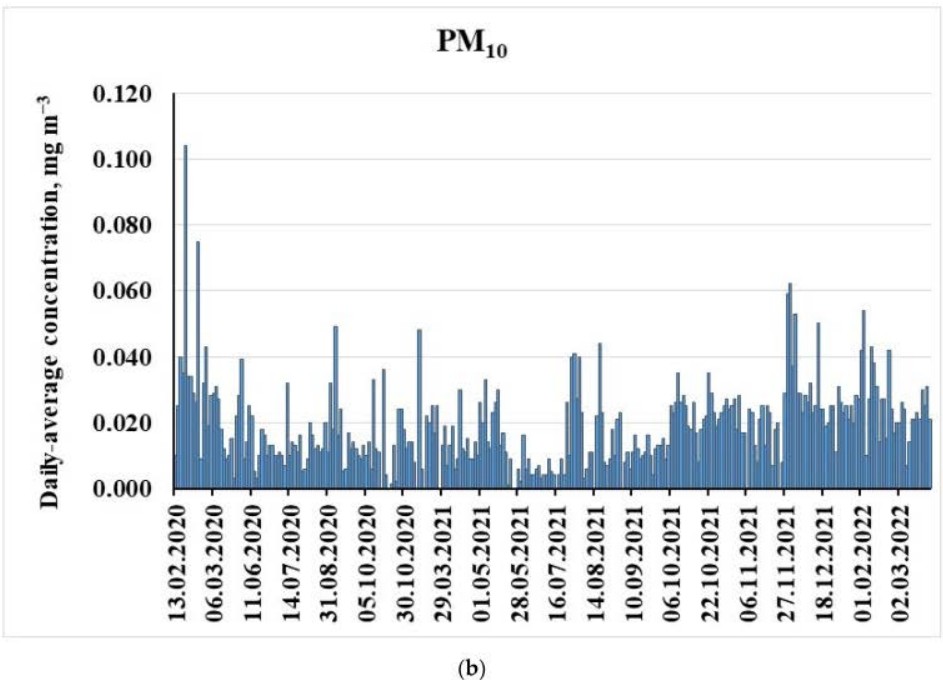

(b)

**Figure 3.** Time series plots of $PM_{2.5}$ (**a**) and $PM_{10}$ (**b**) concentrations in the atmospheric air in Sevastopol.

As it can be seen in Figure 3, concentrations of PM in the air in Sevastopol in 2021–2022 are slightly higher, as compared to previous years. The average ratio $PM_{2.5}$:$PM_{10}$ is approximately 0.7. These data are in agreement with [16], where local $PM_{2.5}$:$PM_{10}$ ratio estimates are between 0.2 and 0.8. Nevertheless, these ratios vary from 0.05 to 0.97 in our study, depending on air masses transported over Sevastopol. The maximum of these ratios were observed from May till August 2021.

### 3.2. Elemental Carbon in the Atmosphere in Sevastopol

In 2021, more than 300 daily air samples for elemental carbon content were analyzed. The concentration of elemental carbon in the air samples collected at the meteorological station did not exceed 0.03 mg m$^{-3}$ (minimal detection limit according to manual [30]). At the same time, there was a slight increase in concentrations since November 2021 to January 2022 (Figure 4), which was probably due to the start and the end of the heating period in the city.

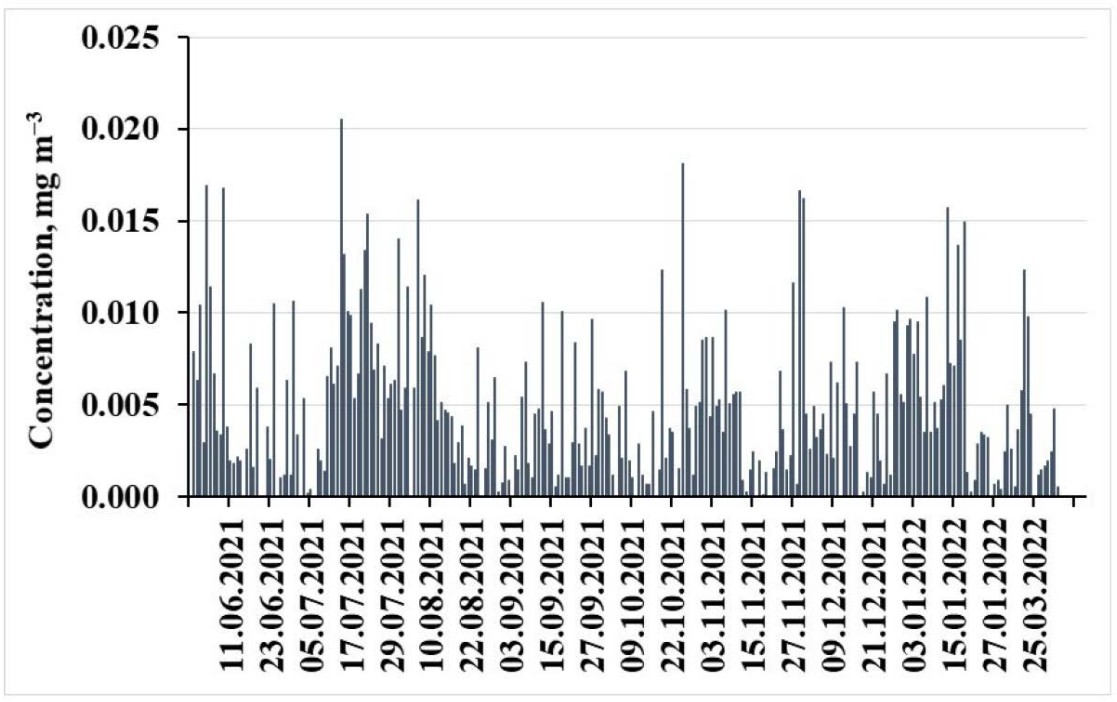

**Figure 4.** Variations of elemental carbon concentrations in the atmosphere in Sevastopol.

Nevertheless, in July 2021 concentrations of elemental carbon in the air were almost 1.5-fold higher than in other periods of research. Dates with increased concentrations of soot matched with dates when increased concentrations of $PM_{2.5}$ were determined (Figure 3). During this month, there were only three cases of precipitation, which, apparently, contributed to the accumulation of elemental carbon and $PM_{2.5}$ in the air.

Taking into account that the meteorological station where air and precipitation samples were collected is located near the sea, it can be considered as a point with a background level of air pollution. It is planned to compare the data obtained during sampling at this meteostation and samples collected in more polluted areas of the city, for example, near highways, existing construction sites, etc. Motorized road transport is one of the dominant sources of urban air pollution in almost all countries [17]. Our preliminary data have shown that elemental carbon concentration in the air samples collected near highways with the traffic intensity of approximately 500–1000 cars per hour exceeds the concentration at the background meteorological station (Figure 1) by 30–50 times, reaching 0.5 mg m$^{-3}$.

### 3.3. Time Series Variation of Silicon and Phosphate Concentrations

During the study period since 2015, more than 500 precipitation samples of atmospheric precipitations have been analyzed. Time series plots of silicon and phosphates concentrations in the atmospheric precipitations from the open sampler are shown in Figure 5.

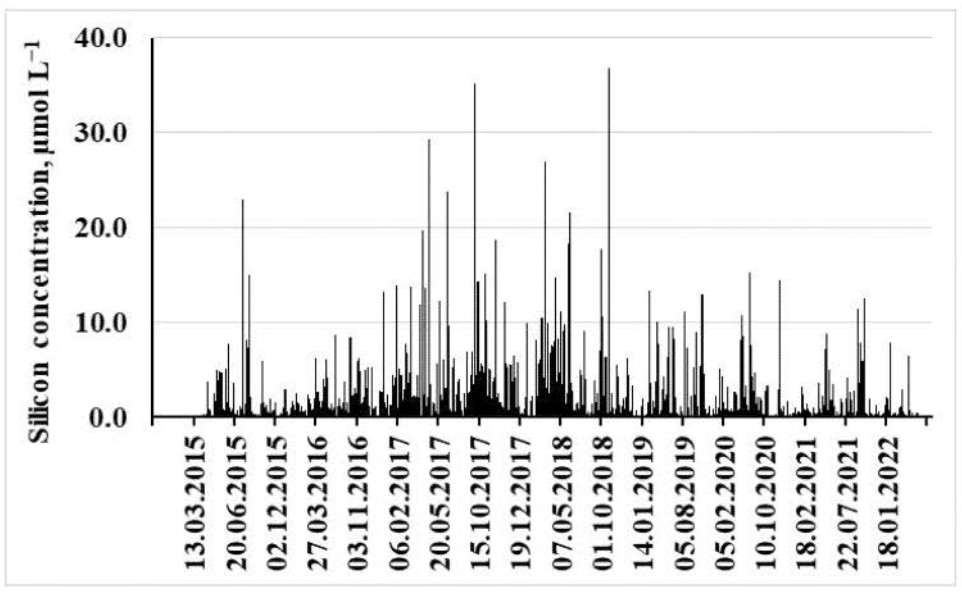

(a)

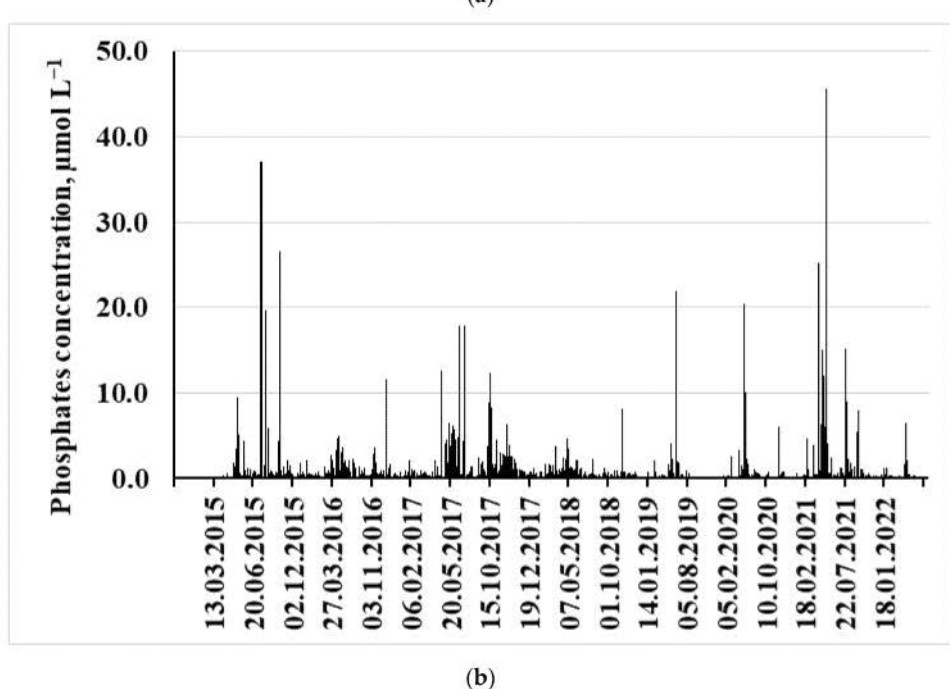

(b)

**Figure 5.** Time series plots of silicon (**a**) and phosphates (**b**) concentrations in the atmospheric precipitations from the open sampler.

The maximum phosphates concentration for 2021 in samples from an open sampler was also determined on 16 May 2021, and it amounted to 45.65 µmol L$^{-1}$, which was more than 26 times higher than the volume-weighted mean (VWM) concentration. In the same sample, the concentrations of silicon exceeded the VWM value for an open sampler by 5-fold.

*3.4. Atmospheric Mid- and Long-Range Transport of $PM_{2.5}$ and $PM_{10}$*

The daily-averaged concentration of $PM_{2.5}$ in the atmosphere of Sevastopol, according to measurements on 15 May 2021, was 0.016 mg m$^{-3}$ (that is 0.5 of MPC), while the $PM_{10}$ concentration was 0.019 mg m$^{-3}$. According to the website https://www.ventusky.com (accessed on 14 June 2022), dust was transported from the southern area of the Black Sea coast during this period. Figure 6 shows the dust cloud over the study region on 15 May 2021.

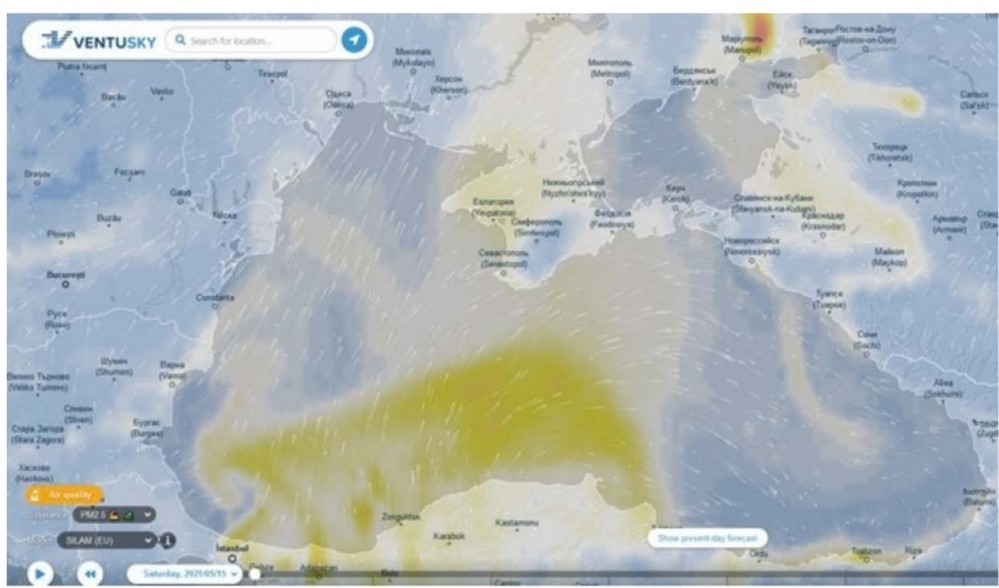

**Figure 6.** Air pollution by $PM_{2.5}$ on 15 May 2021 according to the website https://www.ventusky.com.

The lifetime of aerosols in the atmosphere has been determined by wet and dry deposition [31]. Precipitation contributes to the deposition of microparticles from the air on the underlying surface. In addition, wet deposition is eventually more important locally or regionally, close to aerosols emission sources, whereas dry deposition would be more important in areas far from emission sources. In the atmospheric precipitation, it is possible to determine silicon, as an indicator of the dust transport from distant regions; phosphorus, as an indicator of small particles that appear during the transport of dust storms, grain trans-shipment or during the production of mineral fertilizers; and oxidized nitrogen, as an indicator of the operation of boilers and engines from various types of transportation.

The daily-averaged concentrations of $PM_{2.5}$ and $PM_{10}$ for October 23–24 were 0.019 and 0.037 mg m$^{-3}$, respectively. According to the website https://www.ventusky.com (accessed on 14 June 2022), the transport of dust aerosol over the Black Sea is visible (Figure 7).

In October–November 2021, increased concentrations of nutrients in atmospheric precipitation were also determined. On October 24, when the south-western transport of dust air masses was observed, the concentrations of all nutrients in the samples from the open sampler were increased. The concentration of silicates was 11.46 μmol L$^{-1}$, phosphates— 5.44 μmol L$^{-1}$, and inorganic nitrogen—297 μmol L$^{-1}$. The excess of concentrations over the VWM for this type of sampler was 6.5; 3.2, and 3.7 times for silicates, phosphates, and inorganic nitrogen.

Dust transport was also recorded on November 29 and 30. As can be seen from Figure 8, the transfer of air masses from the African continent was recorded for both days.

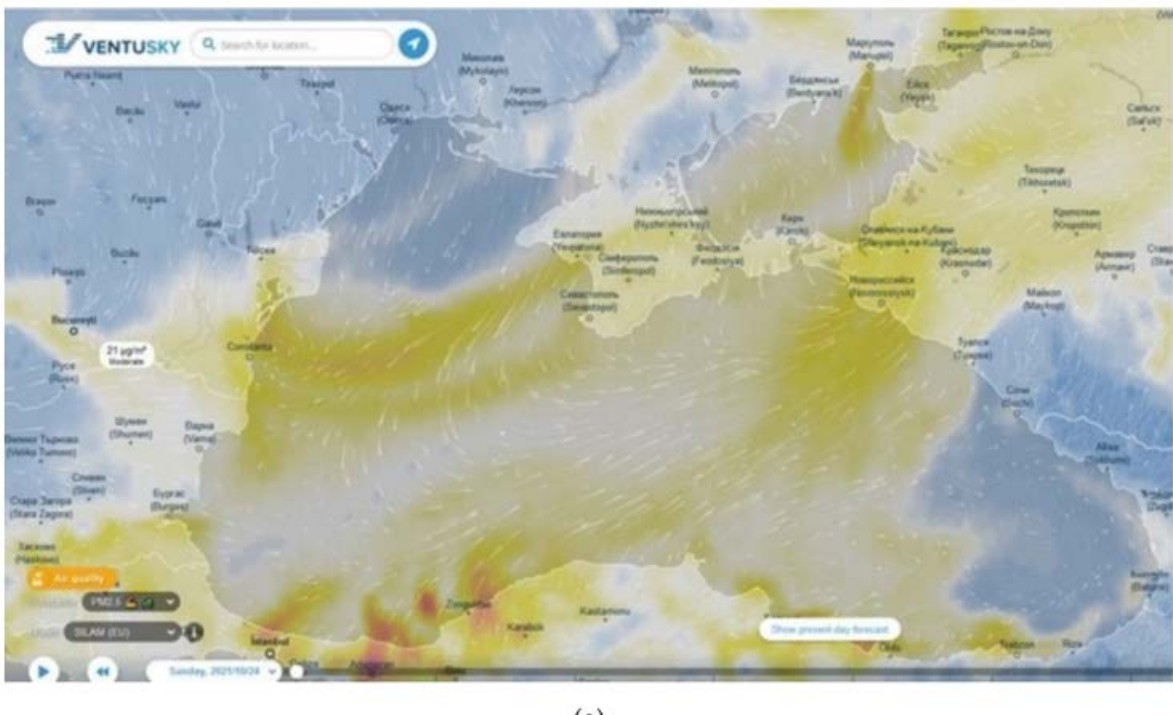

(**a**)

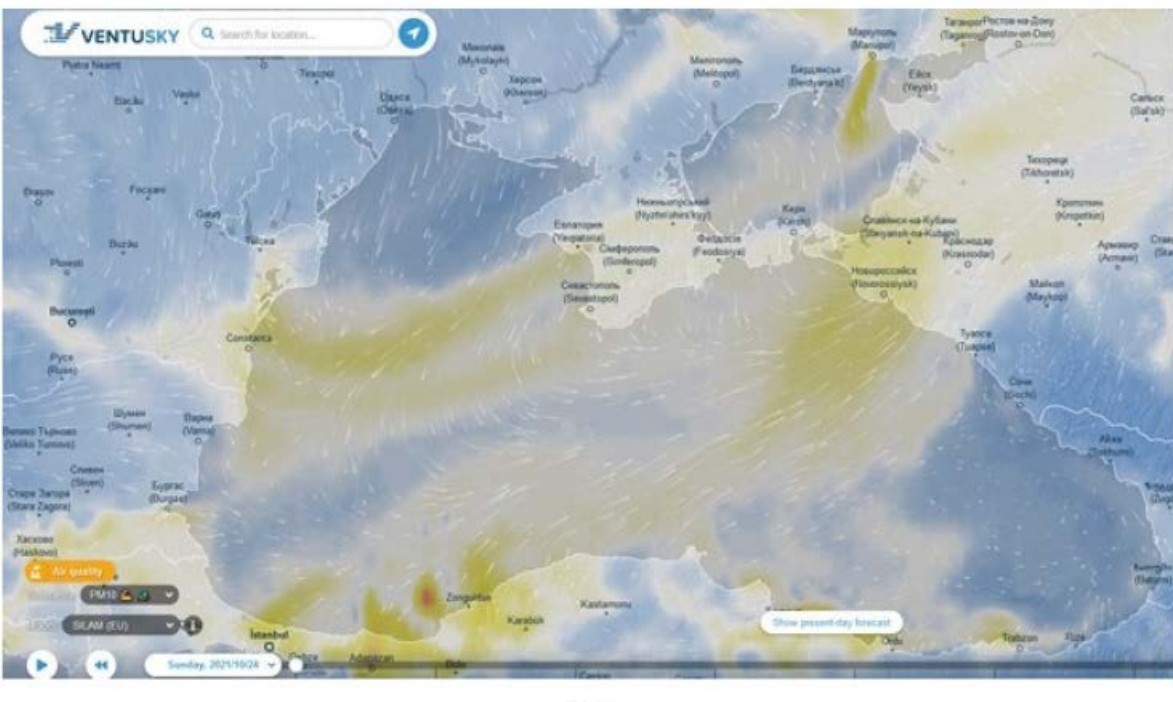

(**b**)

**Figure 7.** Air pollution by PM$_{2.5}$ (**a**) and PM$_{10}$ (**b**) in 24 October 2021 according to the website https://www.ventusky.com (accessed on 14 June 2022).

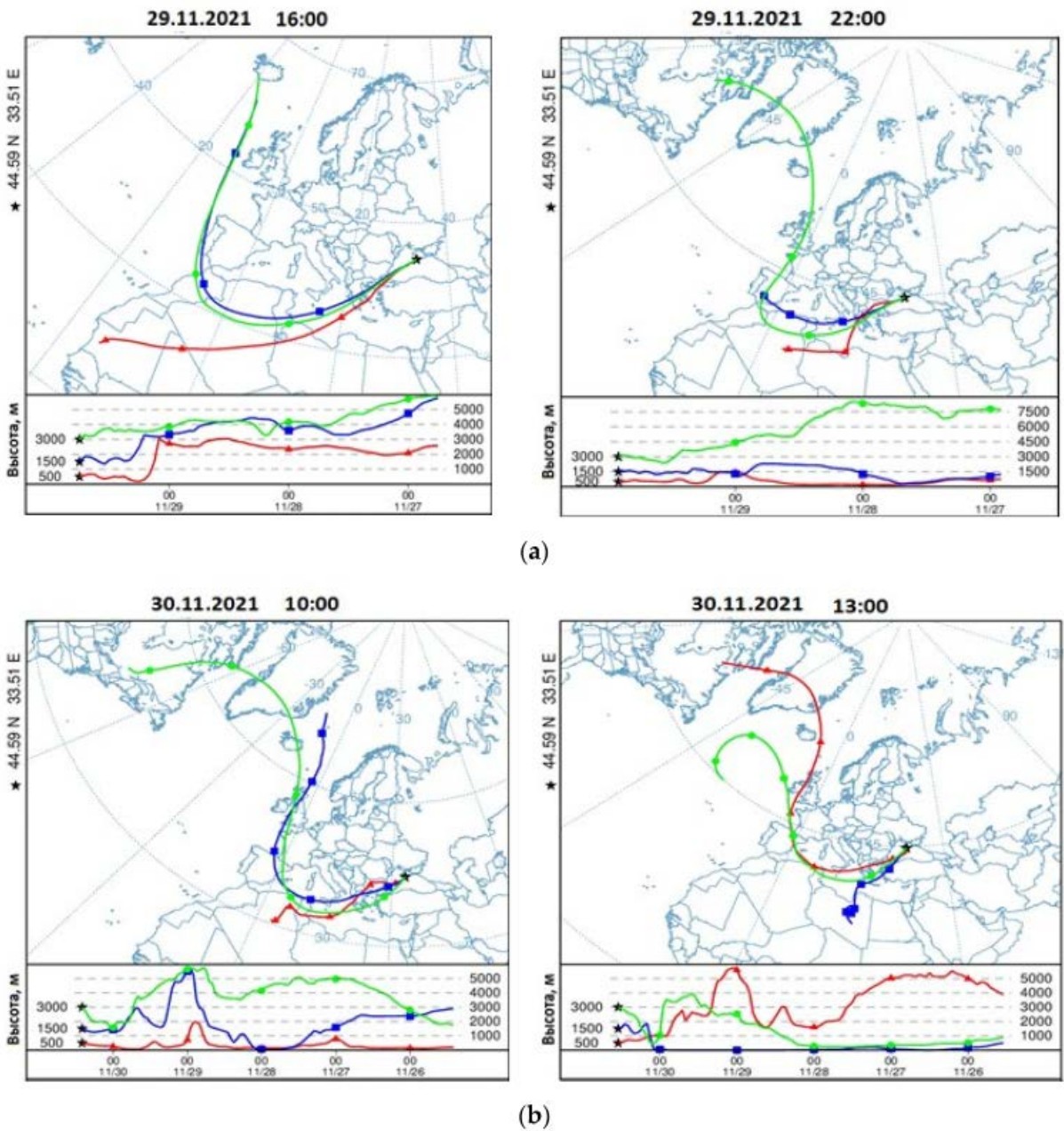

**Figure 8.** Air masses back-trajectories on 29 (**a**) and 30 (**b**) November 2021 (*—location of station Sevastopol).

Data from the website https://www.ventusky.com (accessed on 14 June 2022). For 29 November 2021 (Figure 9) a south-western direction of aerosol transport is also shown.

On 29 November 2021, during the transport of dust air masses over Sevastopol (Figures 8 and 9), light rain was observed. This rain supported precipitation of the dust particles from the atmosphere at the underlying surface. Considering that precipitations washed out dust particles, the concentrations of PM in the atmosphere were not measured. However, concentrations of nutrients in these precipitations were analyzed. The concentration of silicates in the atmospheric depositions collected with the open sampler was 12.43 $\mu$mol L$^{-1}$. This value was more than 7 times higher than the VWM concentration of this element in 2021 for an open sampler and almost 38 times higher than the VWM concentration for a wet-only sampler, which is not affected by dry precipitation. The consequences of the dust air masses transport over the territory of Crimea in the form of settled particles of terrigenous origin on the cars are shown in the photograph (Figure 10). These

photographs were made in different districts of Sevastopol after the end of precipitation on 29 November 2021.

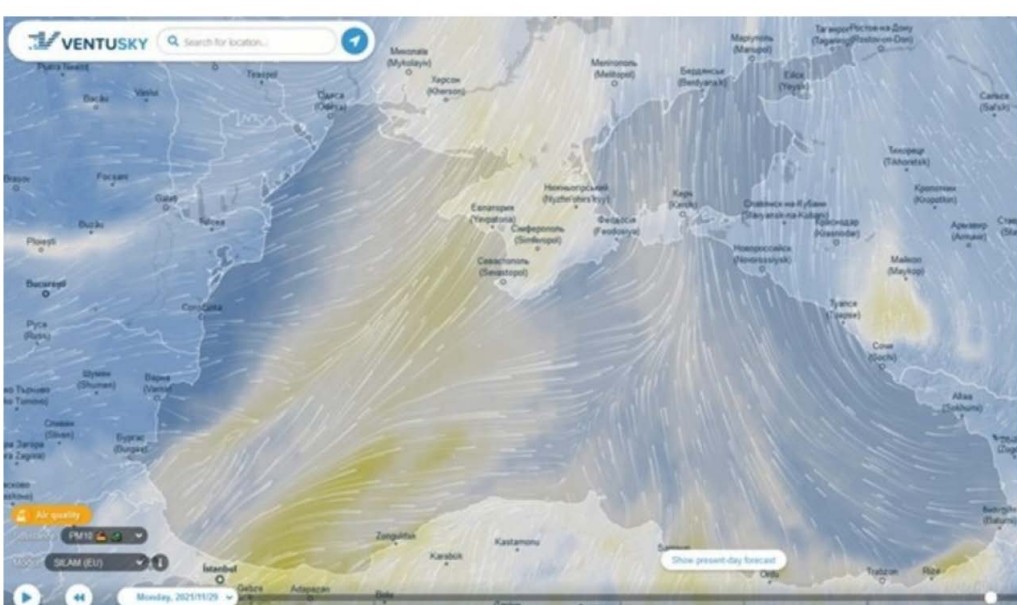

**Figure 9.** Aerosol transport on 29 November 2021 according to the website https://www.ventusky. com (accessed on 14 June 2022).

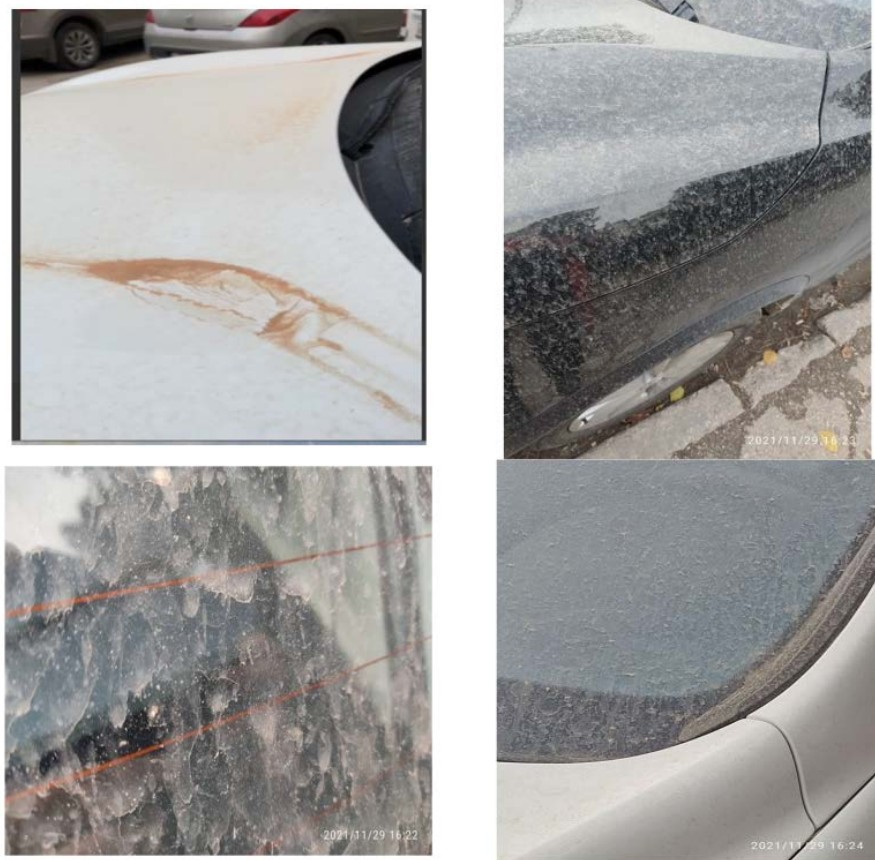

**Figure 10.** Terrigenous particles deposited on cars in Sevastopol as a result of the dust storm from the Sahara on 29 November 2021.

### 3.5. Seasonal Variation of PM$_{2.5}$ and PM$_{10}$

The study of the influence of intra-annual variability of meteorological parameters on the level of pollutants in the air of cities is important not only from a scientific point of view but from a perspective of practical importance. Taking into account the intra-annual variability of meteorological parameters, it is possible to make decisions for carrying out activities that contribute to reducing the level of pollution in the city. The assessment of intra-annual changes in the concentration of fine particles in the air in Sevastopol showed an increase in their concentrations during the cold period of the year (Figure 11).

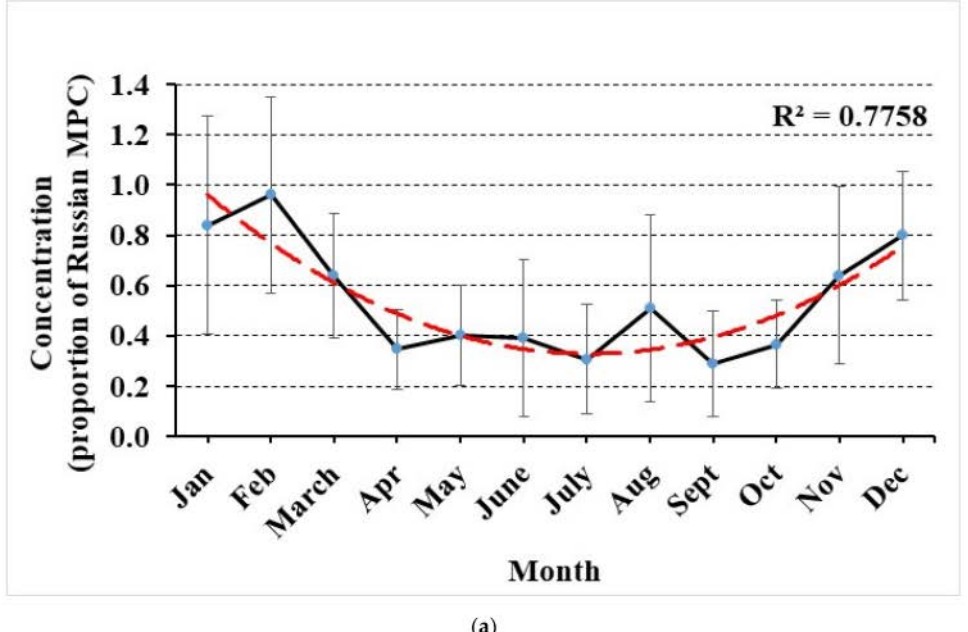

(a)

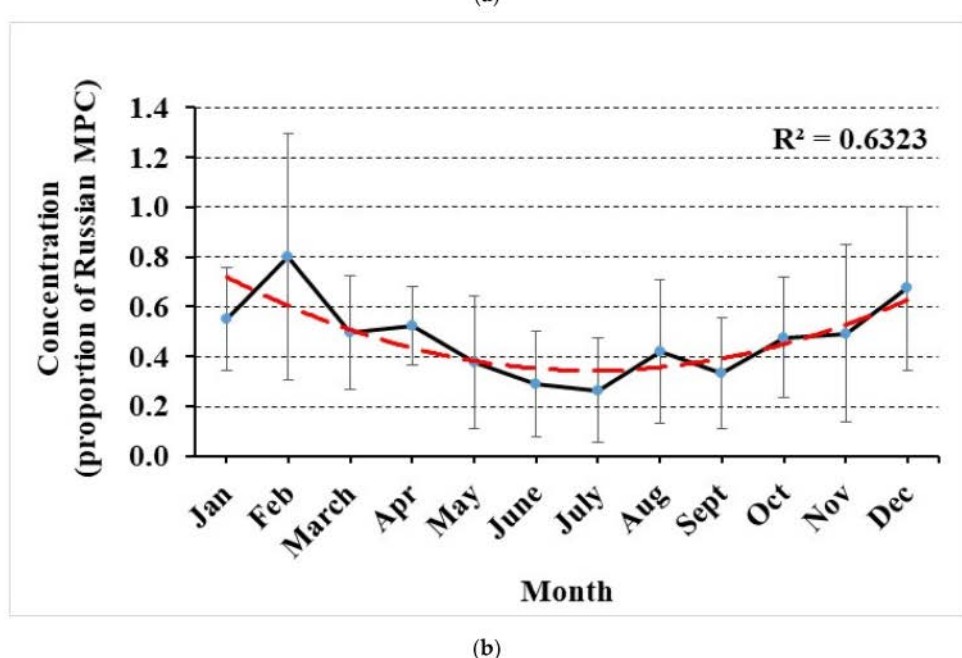

(b)

**Figure 11.** Intra-annual change in concentrations of PM$_{2.5}$ (**a**) and PM$_{10}$ (**b**), expressed in proportion of Russian Federation MPC (the red dotted line is the trend line).

As can be seen from Figure 10, PM concentrations (in proportion of MPC) slightly increase in the cold period of the year. As the weather conditions are fundamentally important for understanding the dynamics of emission, accumulation, and dispersion of

atmospheric polluting substances [32], we have analyzed these results depending on the frequency of days with an atmospheric pressure greater than or equal to 1015 hPa. The most unfavorable for the ecological state of the air basin over the city is the high atmospheric pressure for a long time. During the period of high atmospheric pressure, the most favorable conditions are created for the retention of suspended particles in the air: weak wind, high air temperature and temperature inversions, lack of precipitation, etc. In addition, in these areas of high atmospheric pressure in the summer, a high temperature regime is formed that may initiate additional chemical reactions of pollutants in the atmosphere. Moisture deficiency is also usually associated with increased atmospheric pressure. A characteristic anticyclonic type of sunny weather can cause photochemical smog with a sufficiently large volume of car exhaust gases. Figure 12 shows the change in the concentration of $PM_{10}$ particles and the number of days with high atmospheric pressure during the study period.

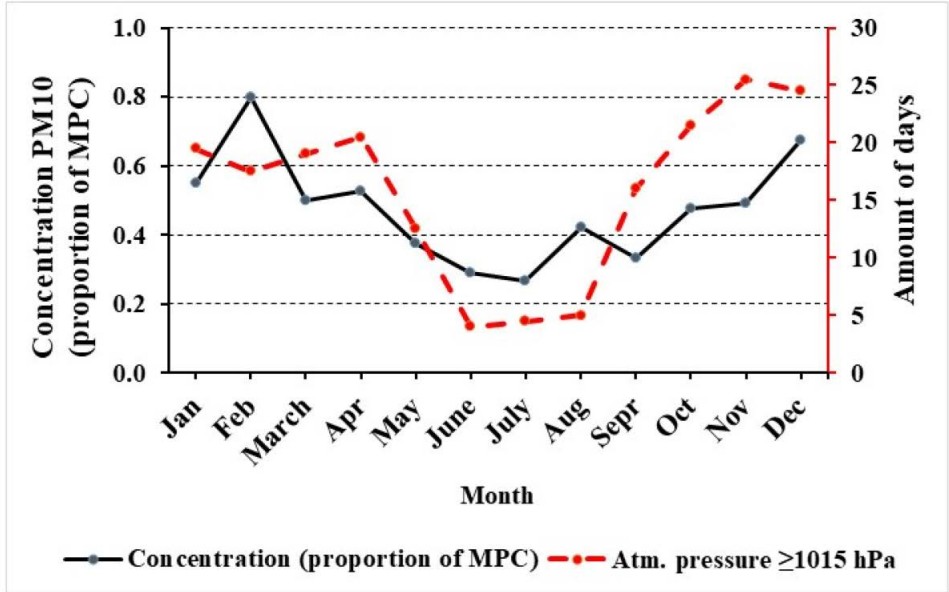

**Figure 12.** The annual variation of the average values of the $PM_{10}$ concentration (in proportion of MPC) and the number of days with atmospheric pressure equal to and above 1015 hPa.

Analysis of the obtained data showed a significant dependence of concentrations of suspended substances in the air on the number of days with high atmospheric pressure. Our data is in agreement with [33–35]. In Russia and the EU countries, the values of the annual and daily-averaged MPC of $PM_{2.5}$ and $PM_{10}$ differ (Table 2).

**Table 2.** MPC standards for the content of $PM_{2.5}$ and $PM_{10}$ in the atmosphere in the Russian Federation and in the EU countries.

|  | $PM_{10}$, mg m$^{-3}$ | | | $PM_{2.5}$, mg m$^{-3}$ | | |
|---|---|---|---|---|---|---|
|  | individual sample | daily-averaged | annual | individual sample | daily-averaged | annual |
| Russia | 0.30 | 0.06 | 0.04 | 0.160 | 0.035 | 0.025 |
| EU |  | 0.05 | 0.02 |  | 0.025 | 0.010 |

Our research shows that in case of dust transport from the deserts of Central Asia, the Sahara, and Syria, the daily-averaged concentrations of $PM_{10}$ and $PM_{2.5}$ particles in the atmosphere of Sevastopol may reach and even exceed the corresponding MPC. The values of the daily-averaged concentrations of microparticles for the entire observation period, compared with the allowed levels adopted in Russia, exceed the MPC values in seven cases for $PM_{2.5}$ and in three cases for $PM_{10}$. The European MPC values have been exceeded in

17 cases for $PM_{2.5}$ particles and in 6 cases for $PM_{10}$. The average annual concentrations of suspended particles (Table 1) are approximately half of the MPC—0.014 mg m$^{-3}$ for $PM_{2.5}$ particles and 0.019 mg m$^{-3}$ for $PM_{10}$ particles, according to Russian standards. However, according to the norms of European countries (Table 2), the average annual concentration of $PM_{2.5}$ even exceeds the established level, and the concentration of $PM_{10}$ corresponds to the maximum permissible concentration.

## 4. Conclusions

The study of air pollution by nutrients, $PM_{2.5}$, $PM_{10}$, and elemental carbon was carried out during 2020–2022.

The results of our study are as follows:

- The maximum phosphates concentration for the samples from an open sampler was more than 26 times higher than the volume-weighted mean concentration; the concentrations of inorganic nitrogen and silicates exceeded the VWM value for an open sampler by 5 times. These concentration values were determined in precipitations after dust air masses transported above the Crimean coast;
- In case of dust transport from the deserts of Central Asia, Sahara, and Syria, the daily-averaged concentrations of $PM_{10}$ and $PM_{2.5}$ particles in the atmosphere of Sevastopol reach and even exceed the corresponding maximum permissible concentrations. The values of the daily-averaged concentrations of microparticles compared with the norms adopted in Russia, exceeded the MPC values in seven cases for $PM_{2.5}$ and in three cases for $PM_{10}$. The European MPC values were exceeded in 17 cases for $PM_{2.5}$ particles and in 6 cases for $PM_{10}$. The average annual concentrations of suspended particles are approximately half of the MPC for $PM_{2.5}$ particles, according to Russian standards. However, according to the norms of European countries, the average annual concentration of $PM_{2.5}$ even exceeds the established level, and the concentration of $PM_{10}$ corresponds to the maximum permissible concentration;
- The duration of the influence of both local sources and long-range atmospheric transport depends on weather conditions: wind speed and direction and atmospheric pressure as well as precipitation, which cleanses the atmosphere of impurities;
- The concentration of soot in the air samples collected at the meteorological station did not exceed the MPC level. However, elemental carbon concentration in the air samples collected near highways exceeds the concentration at the background station by 30–50 times.

**Funding:** This work was carried out within the framework of the state assignments of Marine Hydrophysical Institute of RAS No. 0555-2021-0005 and the Russian Foundation for Basic Research (RFBR) according to research project No. 19-05-50023.

**Institutional Review Board Statement:** Not applicable.

**Informed Consent Statement:** Not applicable.

**Data Availability Statement:** All the data and additional information supporting the findings of this study are available from the corresponding author upon reasonable request.

**Conflicts of Interest:** The author declares no conflict of interest.

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
