# Peer review of "The Characteristics of PM2.5 and PM10 and Elemental Carbon Air Pollution in Sevastopol, Crimean Peninsula"

_applsci, doi:10.3390/app12157758_

Round 1

Reviewer 1 Report

Dear authors,

The paper entitled “Air Pollution Characteristics of PM2.5 and PM10 and Carbon- 2 Containing Aerosol (Soot) in Sevastopol, Russia” deals with the process of formation of atmospheric air pollutants in Sevastopol (Crimea, Russia). The topic of the paper is of interest for the journal “applied sciences” of MDPI editor. The paper is well written and could be accepted after some minor revisions as follows:

-          The introduction is well orginized but i would suggest to enrich the bibliography about other studies of air pollution in urban areas such as:

1.       Caramagna et al. (2015). Analysis of vertical profile of particulates dispersion in function of the aerodynamic diameter at a congested road in Catania, Energy Procedia, 82, 702-707.

2.       Brauer et al. (2016). Ambient Air Pollution Exposure Estimation for the Global Burden of Disease 2013, Environmental Science and Technology, 50(1), 79-88.

3.       Giulia et al. (2015). Urbani air quality management- A review, Atmospheric Pollution Research, 6(2), 286-304.

-          The euquations are not in the same format;

-          The units of measure should me checked and be in the international system (PM2.5 and PM10 for example)

-          The english form is generally good but a deeper proofreading is necessare since many missprints are still present

-          Figures, especially figure 3 and 4, are too little. I would suggest to enlarge them and consider a better quality

Kind Regards

Author Response

Author is thankful for efforts of reviewer, who have contributed their time and efforts to improve the paper. We are thankful for comments and we have accepted almost all of them and made appropriate changes in the text.

Regarding the content:

  1. “The introduction is well orginized but i would suggest to enrich the bibliography about other studies of air pollution in urban areas such as:
  2. Caramagna et al. (2015). Analysis of vertical profile of particulates dispersion in function of the aerodynamic diameter at a congested road in Catania, Energy Procedia, 82, 702-707.
  3. Brauer et al. (2016). Ambient Air Pollution Exposure Estimation for the Global Burden of Disease 2013, Environmental Science and Technology, 50(1), 79-88.
  4. Giulia et al. (2015). Urbani air quality management- A review, Atmospheric Pollution Research, 6(2), 286-304."

Author have added this information in paper.

  1. “The euquations are not in the same format”

I’m sorry, but I do not understand, exactly which equations reviewer meant.

  1. “The units of measure should me checked and be in the international system (PM2.5 and PM10 for example)”

Author have checked the units of measure and corrected.

  1. “Figures, especially figure 3 and 4, are too little. I would suggest to enlarge them and consider a better quality”

Figures had been enlarge to better understanding

  1. “The english form is generally good but a deeper proofreading is necessare since many missprints are still present”

All fixes were made using the "Track Changes" function in Microsoft Word to make them visible. I am planning to contact an English editing service MDPI to improve the language.

Reviewer 2 Report

·     The analysis in the Abstract is mostly qualitative analysis, and the quantitative analysis results are lacking.

·     As stated by the authors in the abstract section, regional air pollution is one of the chronic problems and it is a problem that needs to be solved for global warming, climate change and the life of living things. Accordingly, in the introduction, the authors are expected to include summaries of previous studies in order to compare the results of the current study and reveal its originality. In this respect, the introduction is rather limited and needs to be expanded. For the introduction, it is recommended that the authors also consider the current articles follow;

 “Evaluating effects of the Covid-19 pandemic period on energy consumption and enviro-economic indicators of Turkish road Transportation”, “Influence of COVID-19 on air pollution caused by commercial flights in Turkey”.

·     The exact contribution of the paper is not very clear. Please highlight the clear contribution of the study to the relevant literature.

·     In the material method section, a flowchart of the measurements performed should be presented. What standard, government regulation and/or directive were the measurements made according to? How was the accuracy, validity or timeliness of the measurement results ensured?

·     Figures should be more clear and some of them should be explained in more detail with reasons.

·     How do the authors match the findings of the study with the pictures given in Figure 7 or what kind of evidence do they provide? The authors should provide more detailed explanations related to Figure 7.

·     The authors should discuss by establishing a cause/effect relationship for the findings given in Figure 3- Figure 9. Measurement results made only in periods do not justify the findings and conclusions of the article.

·     Deep and mechanistic discussions are required to explain the results obtained. The section "Results and discussion" is reportorial. A proper comparative analysis of the paper with previous papers should be properly discussed.

·     In the conclusion section, numerical values (i.e., percentages) of the findings of the study should be presented. In addition, the conclusion section is missing some perspectives related to future research work.

Author Response

Author is thankful for efforts of reviewer, who have contributed their time and efforts to improve the paper. We are thankful for comments and we have accepted almost all of them and made appropriate changes in the text.

  1. “The analysis in the Abstract is mostly qualitative analysis, and the quantitative analysis results are lacking.”

Author have added this information in paper.

  1. “As stated by the authors in the abstract section, regional air pollution is one of the chronic problems and it is a problem that needs to be solved for global warming, climate change and the life of living things. Accordingly, in the introduction, the authors are expected to include summaries of previous studies in order to compare the results of the current study and reveal its originality. In this respect, the introduction is rather limited and needs to be expanded. For the introduction, it is recommended that the authors also consider the current articles follow;

 “Evaluating effects of the Covid-19 pandemic period on energy consumption and enviro-economic indicators of Turkish road Transportation”, “Influence of COVID-19 on air pollution caused by commercial flights in Turkey”.”

Author have changed an Introduction and have added information in paper

  1. “In the material method section, a flowchart of the measurements performed should be presented. What standard, government regulation and/or directive were the measurements made according to? How was the accuracy, validity or timeliness of the measurement results ensured?”

Author have added an information about Methods and directives.

  1. “Figures should be more clear and some of them should be explained in more detail with reasons.

“How do the authors match the findings of the study with the pictures given in Figure 7 or what kind of evidence do they provide? The authors should provide more detailed explanations related to Figure 7.”

“The authors should discuss by establishing a cause/effect relationship for the findings given in Figure 3- Figure 9. Measurement results made only in periods do not justify the findings and conclusions of the article.”

Figures had been enlarge to better understanding. Author tried to explain results and connections for Figures more clearly.

  1. “Deep and mechanistic discussions are required to explain the results obtained. The section "Results and discussion" is reportorial. A proper comparative analysis of the paper with previous papers should be properly discussed.”

Author have added comparative analysis of the paper with previous results published by other researchers.

  1. “In the conclusion section, numerical values (i.e., percentages) of the findings of the study should be presented. In addition, the conclusion section is missing some perspectives related to future research work.”

 Author have added this information

All fixes were made using the "Track Changes" function in Microsoft Word to make them visible. I am planning to contact an English editing service MDPI to improve the language.

Round 2

Reviewer 2 Report

The authors have substantially improved their manuscript and answered most of my previous concerns. Its technical and scientific value have improved fairly. English grammar errors are corrected. In my opinion, it now has the quality to be published in the Applied Sciences as is.

I accept the publication of the revised manuscript applsci-1794116 in the Applied Sciences.

Author Response

Thank you very much for your comments and suggestions.